# CRISPR/Cas9-Mediated Multiple Knockouts in Abscisic Acid Receptor Genes Reduced the Sensitivity to ABA during Soybean Seed Germination

**DOI:** 10.3390/ijms232416173

**Published:** 2022-12-18

**Authors:** Zhaohan Zhang, Wanpeng Wang, Shahid Ali, Xiao Luo, Linan Xie

**Affiliations:** 1Key Laboratory of Saline-Alkali Vegetative Ecology Restoration, Ministry of Education, College of Life Science, Northeast Forestry University, Harbin 150040, China; 2Peking University Institute of Advanced Agricultural Sciences, Weifang 261325, China; 3College of Life Sciences and Oceanography, Shenzhen University, Shenzhen 518060, China

**Keywords:** ABA, CRISPR/Cas9, GmPYLs, seed germination, soybean

## Abstract

Abscisic acid (ABA) is an important plant hormone that regulates numerous functions in plant growth, development, and stress responses. Several proteins regulate the ABA signal transduction mechanism in response to environmental stress. Among them, the PYR1/PYL/RCAR family act as ABA receptors. This study used the CRISPR/Cas9 gene-editing system with a single gRNA to knock out three soybean PYL genes: *GmPYL17*, *GmPYL18*, and *GmPYL19*. The gRNA may efficiently cause varying degrees of deletion of *GmPYL17*, *GmPYL18*, and *GmPYL19* gene target sequences, according to the genotyping results of T0 plants. A subset of induced alleles was successfully transferred to progeny. In the T2 generation, we obtained double and triple mutant genotypes. At the seed germination stage, CRISPR/Cas9-created *GmPYL* gene knockout mutants, particularly *gmpyl17/19* double mutants, are less susceptible to ABA than the wild type. RNA-Seq was used to investigate the differentially expressed genes related to the ABA response from germinated seedlings under diverse treatments using three biological replicates. The *gmpyl17/19-1* double mutant was less susceptible to ABA during seed germination, and mutant plant height and branch number were higher than the wild type. Under ABA stress, the GO enrichment analysis showed that certain positive germination regulators were activated, which reduced ABA sensitivity and enhanced seed germination. This research gives a theoretical basis for a better understanding of the ABA signaling pathway and the participation of the key component at their molecular level, which helps enhance soybean abiotic stress tolerance. Furthermore, this research will aid breeders in regulating and improving soybean production and quality under various stress conditions.

## 1. Introduction

Soybean (*Glycine max*) is an important economic crop; it has high nutritional value and is commonly used as a prime source of vegetable oils and proteins for human consumption [1,2,3]. In many plants, especially annual seed plants, seed germination plays a key role in the survival and spread of plants under suitable environmental conditions [4,5].

Abscisic acid (ABA) is an essential plant hormone; which governs various aspects of plant growth and development, including seed development, bud dormancy, and is a negative regulator of seed germination [6,7,8,9]. There is an antagonistic relationship between dormancy and seed germination of plants [10]. During germination, the ABA content dropped, indicating that the ABA signaling pathway was inhibited. The seeds of typical ABA defective mutants germinated faster than the wild types [11]. The seeds of transgenic plants constitutively expressing ABA biosynthetic genes maintained deep dormancy [12,13]. However, the high accumulation of ABA in ABA catabolism mutants led to the high dormancy of seeds [14]. Seed germination is affected by the biosynthesis of ABA and the signal transduction-dependent pathway of ABA [15]. The seeds of *abi1* and *abi2* (ABA insensitive mutant) showed that sensitivity to ABA inhibition decreased during germination [16,17]. The transcription factor DREB2C (Dehydration-responsive element-binding proteins 2C) participates as a positive regulator of ABA biosynthesis during seed germination; overexpression of DREB2C conferred ABA hypersensitivity during germination [18]. HONSU belongs to a group A PP2C family, which negatively regulates seed dormancy by inhibiting ABA signaling. HONSU mRNA expression levels were inversely correlated with the germinability of seeds [19]. The AHG3 (ABA-HYPERSENSITIVE GERMINATION 3) belongs to a clade of PP2Cs. Under ABA treatment, the seed germination rate of the *ahg3* mutant was decreased [20,21].

ABA signaling is a complex pathway that is regulated by different factors. The ABRE (ABA-responsive elements) are the major *cis*-acting elements in the ABA-responsive gene expression [22,23]. ABF (ABRE-binding factor) regulates ABRE-dependent gene expression [24]. SNF1-associated protein kinase 2 (SnRK2) is a key regulator of ABA signal, including AREB/ABF regulator, ABA receptor, and type 2C protein phosphatase (PP2Cs) [25]. The PYLs is an ABA receptor located at the top of the abscisic acid negative regulatory pathway, with negative regulatory factor PP2C and positive regulatory factor SnRK2, and controls ABA signal transduction [26,27]. In *Arabidopsis thaliana*, the exogenous application of ABA altered 11 out of 14 *PYLs* genes. PYL11 and PYL12 are only found in mature seeds and have a beneficial effect on ABA-mediated seed germination [28]. However, the potential regulatory mechanism of PYLs on soybean seed germination is unclear.

The Clustered Regularly Interspaced Short Palindromic Repeats (CRISPR)-Cas9 system is a simple, efficient and precise soybean genome editing tool [29,30,31]. The CRISPR/Cas9 genome editing system consists of two essential functional components: the Cas9 endonuclease and the guide RNA (gRNA), which binds to a particular target DNA sequence that ends with a short DNA sequence known as the protospacer adjacent motif (PAM) [32]. CRISPR/Cas9 nuclease-based gene drives may generate a targeted DNA double-strand break (DSB) in a genome when used for genome editing. Non-homologous end joining (NHEJ) may produce insertion/deletion (indel) mutations, gene replacements, and single base pair conversions at the break site [33,34]. In recent years, multiple editing genes have been successfully applied to the functional identification of soybean genes and germplasm resources [35,36,37,38,39]. This study utilized the CRISPR/Cas9 technology to induce premature translational termination of *GmPYL17*, *GmPYL18*, and *GmPYL19*, three homologous genes encoding PYL protein, to investigate the role of PYL in soybean seed germination.

Results revealed that premature translational termination of *GmPYL17*, *GmPYL18*, and *GmPYL19* influenced soybean germination under ABA stress using T2 generation homozygous mutants. Compared with single amino acid premature translational termination, double amino acid premature translational termination showed a more significant increase in seed germination rate. Furthermore, the mutants enhanced the plant height and branch number more substantially than the wild type. These results confirmed that *GmPYL17*, *GmPYL18*, and *GmPYL19* genes play an important role in seed germination and ABA signal transduction, which provides a basis for further understanding the mechanism of the ABA core signal pathway in soybean.

## 2. Results

### 2.1. GmPYL17, GmPYL18, and GmPYL19 Are Homologous with AtPYL8 and Expressed Specifically in Dry Seeds

Protein sequence alignment was used to identify PYLs protein homologs in legumes with other plant species. After amino acid sequence alignment and verification on the NCBI database BLASTp, three genes in soybean, two in alfalfa, one in common bean, four in rice, and four in tomato were revealed to be homologous to AtPYL8 (At5g53160). In phylogenetic tree construction, the *Marchantia polymorpha* is the root node, the common ancestor of all the species in the tree. Three soybean homolog genes *Glyma.13G041800*, *Glyma.03G066200*, and *Glyma.06G126100*, were named *GmPYL17*, *GmPYL18*, and *GmPYL19* and displayed homology to *AtPYL8* from *Arabidopsis*. The 15 homolog genes with *AtPYL8* clustered into two groups, dicotyledons, and monocotyledons. According to the phylogenetic analysis, legumes were grouped within the dicotyledonous group, and these three soybean PYL8 proteins were closely related to those of alfalfa and kidney beans (Figure 1A–C). These PYL proteins contain a common polyketide cyclase domain 2 (Pfam: Polyketide_cyc2). It is a member of the superfamily of the START domain and the key domain of PYL as an ABA receptor [40]. There are six conserved motifs (motif1~motif6) which were identified in six species, with lengths of 50, 50, 50, 21, 11, and 29 amino acids, respectively (Appendix A). The motif shared by all PYLs proteins is motif1, motif2, and motif3, which are consistent with the multiple sequence alignment analysis (Figure 1D and Appendix A). The FPKM level of *GmPYL17*, *GmPYL18*, and *GmPYL19* genes in seeds was significantly higher than that in aboveground parts, suggesting that these three genes may play an important role in seeds (Figure 1E). To explore the function of *GmPYL17*, *GmPYL18*, and *GmPYL19* in seeds, we used the CRISPR-Cas9 approach to create the loss of function mutants of these three genes in Dongnong 50 (DN50) background.

### 2.2. Generation of Transgenic Soybean Plants Harboring the CRISPR/Cas9 Expression Module

We used CRISPR-cas9 to create multiple mutants in three lines. In the T_2_ generation, the homozygous mutant materials were screened. Among them, five bases were deleted at 1700–1705 bp of the *GmPYL17* gene in *gmpyl17/19-1* mutant line and two bases at 348–350 bp of the *GmPYL19* gene. In the *gmpyl17/18/19-1* mutant line, five bases at 1700–1705 bp of the *GmPYL17* gene were deleted, seven bases at 1676–1683 bp of the *GmPYL18* gene were deleted, and three bases at 347–350 bp of the *GmPYL19* gene were deleted. *gmpyl17/18/19-2* mutant line has five bases deleted at 1700–1705 bp of *GmPYL17* gene, four bases deleted at 1680–1684 bp of *GmPYL18* gene, and nine bases deleted at 342-351 bp of *GmPYL19* gene. (Figure 2A–C).

The homozygous mutant genes that have been sequenced were further verified by cleaved amplified polymorphism sequences (CAPS). The PCR products of wild type, heterozygous mutant, and homozygous mutant were digested by restriction endonuclease according to different mutation types of different genes (Appendix A). The results showed homozygous mutations in all mutant lines (Figure 3A–C). The homozygous mutant of the *GmPYL17* gene was cut into two bands after *PvuI* digestion, the heterozygous mutant had three bands, and the wild type could not be cut into a single band. The homozygous mutant of the *GmPYL18* gene could not be cut into one band after *SmII* digestion, but the wild type could be cut into two bands, while the homozygous mutant of the *GmPYL19* gene could not be cut into one band after *SmII* digestion (the size of the band was similar) (Figure 3D).

The deduced protein sequence was analyzed according to the sequencing results of homozygous T_2_ transgenic lines. Base alterations cause frameshift changes in all mutant lines, resulting in premature translation termination or amino acid deletions. The frameshift mutation of the *GmPYL17* protein results in the premature termination of glutamate at position 81 in the *gmpyl17/19-1* mutant line, while the frameshift mutation of *GmPYL19* protein results in the premature termination of valine at position 74 in the *gmpyl17/19-1* mutant line. The frameshift mutation of the *GmPYL17* protein results in the early termination of glutamate at position 81, the frameshift mutation of the *GmPYL18* protein results in the early termination of glycine at position 79, and the frameshift mutation of *GmPYL19* protein results in glutamate deletion at position 67 in the *gmpyl17/18/19-1* mutant line. The frameshift mutation of *GmPYL17* leads to the early termination of glutamate at position 81 in *gmpyl17/18/19-2* mutant lines, and the frameshift mutation of *GmPYL18* leads to the early termination of serine at position 80 in *gmpyl18/19-2* mutant lines, and the frameshift mutation of *GmPYL19* leads to the deletion of aspartic acid, leucine, and glutamate at position 65–67 in *gmpyl19* (Figure 4A–C).

### 2.3. The gmpyl17/19-1 Mutant Has Decreased ABA Sensitivity in Seed Germination

The MS medium with 0 μM (CK) and 10μM ABA was used to analyze the germination rate of *gmpyl17/19-1*, *gmpyl17/18/19-1* mutant, and wild type. We define the seed hypocotyl to break through the seed coat as the boundary to determine whether the seed germinates or not. After 36 h, there was no significant difference in the seed germination rate of WT, *gmpyl17/19-1,* and *gmpyl17/18/19-1* mutant treated with 0 μM (Figure 5A,B). However, under the treatment of 10 μM ABA, the seed germination rate of *gmpyl17/19-1* mutant was significantly higher than that of the wild type. The root length was also significantly higher than that of WT (Figure 5C). After 72 h, the germination rate of WT, *gmpyl17/19-1,* and *gmpyl17/18/19-1* mutant treated with 0 μM reached 100%. However, under the treatment of 10 μM ABA, the germination rate of *gmpyl17/19-1* mutant was close to 80%, and the germination rate of WT was less than 50%. The root length of *gmpyl17/19-1* mutant was still significantly higher than that of WT (Figure 5D,E). These indicated that 10μM ABA inhibited the seed germination of WT and *gmpyl17/18/19-1* mutant but had less effect on the *gmpyl17/19-1* mutant. We previously found that the *gmpyl17/18/19-2* mutant also showed that seed germination was inhibited by ABA, which was similar to that of the *gmpyl17/18/19-1* mutant (not shown). According to the amino acid sequence analysis, we found that the *GmPYL19* gene of *gmpyl17/19-1* mutant was terminated prematurely, and the amino acid deletion of the *GmPYL19* gene occurred in *gmpyl17/18/19-1* and *gmpyl17/18/19-2* mutants (Figure 4C). Therefore, we think the *GmPYL19* gene may be more critical in responding to ABA stress during seed germination.

### 2.4. The gmpyl17/19-1 Mutant Induce Positive Regulators of Seed Germination

The sensitivity to ABA was reduced during seed germination due to *GmPYL17* and *GmPYL19* gene mutations. The RNA-seq was performed from the mutant seeds of *gmpyl17/19-1* line after 36 h of treatment with 10 μM ABA and control. When the differentially expressed genes (DEGs) were calculated (absolute log2 fold value > 1 and adjusted *p* < 0.05), the *gmpyl17/19-1* mutant had fewer DEGs (5250 DEGs, with 1515 upregulated and 3735 downregulated) compared to WT (10,369 DEGs, with 4733 upregulated and 5636 downregulated) in response to ABA stress. A lesser proportion of ABA stress-repressed DEGs (71.14%) in the *gmpyl17/19-1* mutant than the 54.35% in WT, whereas the upregulated DEGs showed a reverse change trend (Figure 6A). Under normal conditions, 2195 DEGs had higher expression levels in the *gmpyl17/19-1* mutant than WT, and 2033 DEGs had lower expression levels in the *gmpyl17/19-1* mutant than WT; and when subjected to ABA stress treatment, more DEGs were present between the genotypes (Figure 6B).

Based on the expression patterns, the DEGs associated with both ABA stress and *GmPYL17*/19 regulation can be grouped into six clades (Figure 6C,D). To test which clade contributes more to the different growth and stress response of the *gmpyl17/19-1* mutant and WT, we performed a Gene Ontology (GO) enrichment analysis of the six clade DEGs (Appendix A, Appendix A). Among them, *Glyma.18g035000* (homologous gene of *AHG3* (Appendix A) and *Glyma.19g069200* (homologous gene of *HONSU* (Appendix A) were included in clade 3. Previous studies have reported that these two genes are positive regulators of seed germination. We knocked out *GmPYL17* and *GmPYL19*, and under ABA treatment, the expression of *GmAHG3* and *GmHONSU* was upregulated, and the seed germination rate increased, which was consistent with the previous research conclusions.

The qRT-PCR results showed no significant difference in the expression level of the *AHG3* gene between WT and *gmpyl17/19-1* mutant without ABA treatment, as did HONSU. However, it is worth noting that neither *AHG3* nor *HONSU* genes were significantly changed in WT after 10 μM ABA treatment, but their expression was significantly increased in *gmpyl17/19-1* mutant (Figure 6E,F). These results indicated that the *gmpyl17/19-1* mutant promoted *AHG3* and *HONSU* expression under 10 μM ABA treatment and improved the germination rate of soybean seeds. Primers used for qRT-PCR in this study are shown in Appendix A.

### 2.5. gmpyl17/19-1 Mutant Enhanced Morphological Traits but Inconsequential Effect on 100-Seeds Weight

To further explore the agronomic characteristics of the mutant lines, the plants were sown in the greenhouse until the flowering stage, the number of branches and plant height were recorded, as well as the 100-seed weight at maturity. At the R2 flowering stage, the number of branches in the mutant lines was significantly higher than in the wild type (Figure 7A–C). The plant height of the R8 mature wild type and mutant was measured, and it was discovered that all mutants were taller than the wild type, with the *gmpyl17/19-1* mutant being substantially higher than the wild type. There was no substantial difference between the mutant and the wild type in terms of 100-seeds weight (Figure 7D).

## 3. Discussion

Soybean provides more than half of the global oilseed production and a quarter of the protein for human food. Soybean seeds contain 40% protein and 20% oil [41]. To guarantee a sufficient food supply for the human population, its source should be increased by 50% by 2050 [42,43,44]. In turn, more effective plant breeding of soybean varieties is urgently needed. A comprehensive evaluation and utilization of genetically diverse germplasm are essential for crop improvement. CRISPR/Cas9 technology has the advantages of complete knockout expression, high specificity, low miss rate, heredity, functional research, etc. It has been successfully used for gene editing in plant species, including Arabidopsis, tomato, rice, wheat, and soybean [45,46,47,48,49,50,51].

In the construction of CRISPR/Cas9 multi-gene editing vector, the design principles of sgRNA still follow the following points: the length of sgRNA is about 20 bp, neither too long nor too short, too long can easily lead to vector construction failure, too short can easily increase the probability of miss events; the position of sgRNA should be before the PAM (NGG) sequence; the GC content of sgRNA should be about 50% [52,53]. However, the experiment shows that the GC content of 70% is also effective; in the design of sgRNA, there is a 1–2 base mismatch far away from the PAM sequence, which can also successfully target genes [54,55]. We also counted the transformation efficiency. Concerning the design of a vector with multiple genes corresponding to a single gRNA, it was found that the efficiency of simultaneous editing of three genes was 3.3%. Although the efficiency of triple editing was relatively low, it was still feasible (Appendix A).

Most flowering plants reproduce through sexual reproduction and seed production. Seed germination not only determines the reproduction and survival of plant populations but also determines the time for plants to enter the natural and agroecosystem, which directly affects the yield of crops. As an important plant hormone, ABA plays an important role in seed germination [28].

Previous studies have shown that rice *PYLs* multi-gene editing materials edited by CRISPR/Cas9 system are not sensitive to ABA during seed germination [47]. Among the rice Subclass-I and -II SnRK2s, osmotic stress/ABA–activated protein kinase 2 (SAPK2) may be the primary mediator of ABA signaling [56]. Rice mutants with *OsSARK2* deficiency also showed low sensitivity to ABA during seed germination [57]. This study also showed that the *gmpyl17/19-1* mutant was less sensitive to ABA than the wild type under the 10 μM ABA treatment. The GO enrichment analysis of the clade1 also enriched many transcription factors, such as Glyma.07G156200 (DREB2C), which regulates seed germination and responds to drought stress [18]. Previous studies have carried out a lot of research on PYL-mediated drought resistance. *AtPYL5* can reduce the transpiration rate of plants under drought stress, minimize water loss, reduce oxidative stress damage, and improve photosynthesis efficiency [58]. PYL4 significantly reduced the water consumption of transgenic wheat throughout its life cycle and maintained wheat yield under drought conditions [59]. Ectopic expression of *AtPYL13* in *Arabidopsis thaliana* leads to ABA hypersensitivity; affects seed germination and stomatal closure [60]. *AtPYL4* can reduce stomatal conductivity and improve water use efficiency [61]. *AtPYR1*, *AtPYL1*, *AtPYL2*, and *AtPYL3* can improve the drought resistance and water use efficiency of *Arabidopsis thaliana*. *AtPYL5* enhances the drought resistance of plants by directly inhibiting type A PP2Cs and activating the ABA signal [62]. The mutants we created may also affect soybean drought or flooding stress, and further experiments are needed to explore these functions.

In the *gmpyl17/19-1* mutant, the mutation type of the *GmPYL19* allele is early termination of translation, but in *gmpyl17/18/19-1* mutant, the mutation type of *GmPYL19* allele is amino acid deletion. Therefore, we believe *GmPYL19* may play a key role in soybean seed germination. In *gmpyl17/19-1* mutant, the mutation type of the *GmPYL19* allele is early termination of translation, whereas, in *gmpyl17/18/19-1* mutant, the mutation type of *GmPYL19* allele is an amino acid deletion. However, the agronomic characters and the mutant lines with early termination of translation during seed germination showed greater differences than those of the wild type, indicating that the early termination of the *GmPYL19* gene may be involved in more regulation of soybean growth, development, and germination. Therefore, *GmPYL17, 18, 19,* and especially *GmPYL19* genes play a key role in seed germination and affect agronomic traits such as plant height and the number of branches. Our results reveal an important new component in soybean seed germination.

## 4. Materials and Methods

### 4.1. CRISPR/Cas9 Vector Construction and Target Selection

The nucleotide sequence of 14 *AtPYLs* and their corresponding encoded proteins were searched from the TAIR database (https://www.arabidopsis.org/, accessed on 17 December 2022). According to the *AtPYLs* amino acid sequence of *Arabidopsis thaliana*, the homologous genes of other species were found from the JGI database (https://phytozome-next.jgi.doe.gov/, accessed on 17 December 2022), the candidate *GmPYLs* proteins were selected.

The NCBI database identified the members of the *GmPYLs* gene family by BLASTp comparison. The amino acid sequences of proteins encoded by members of the *OsPYLs* gene family were obtained from the NCBI database (http://plants.ensembl.org/index.html, accessed on 17 December 2022). The phylogenetic tree was constructed using the Neighbor-joining method by MEGA 10, and 3000 confidence tests were carried out by the bootstrap method to analyze the phylogenetic relationship between the *GmPYLs* gene family and *AtPYLs* and *OsPYLs* gene family [63]. Finally, the homozygous genes with the *AtPYL8* gene in soybean were determined.

The skeleton of the vector is pSC1, sgRNA was promoted by the GmU6 promoter, and the Cas9 gene was expressed by the GmUbi3 promoter (provided by Yu Deyue, Nanjing Agricultural University); *Escherichia coli*, DH5α (purchased from Transgen Biotech Company); *Agrobacterium tumefaciens*, EHA105 (Provided by Feng Xianzhong, Northeast Institute of Geography and Agricultural Ecology, Chinese Academy of Sciences). The soybean cultivar used in the experiment was Dongnong50 (DN50) (Provided by Jiang Zhenfeng, Northeast Agricultural University). DNA oligonucleotides and gRNA are provided by Qingke Biology Company (Appendix A). LgUI digested the empty pSC1 vector; using T4 ligase, the digested vector was ligated with dimerized oligonucleotides, and finally, the transgenic vector was constructed. Sanger sequencing was performed to validate the transgenic vector’s sequences and orientation of different components. The constructed plasmid was transferred into *Agrobacterium tumefaciens* EHA105 for the stable transformation of the soybean.

### 4.2. Stable Soybean Transformation and Transgene Confirmation

The cotyledon *Agrobacterium tumefaciens*-mediated genetic transformation was carried out according to the established procedure, and the recipient was Dongnong 50 soybean variety [64]. When the first triple compound leaf grew out of the tissue culture seedling (V1 stage), a compound leaf was cut, and the DNA of soybean leaves was extracted by the CTAB method (Appendix A). The PCR reaction was carried out with the specific primers of the pSC1-Cas9 gene and gRNA, which was used to verify the successful transfer of the vector into the tissue culture plant.

### 4.3. Acquisition and Identification of GmPYLs Mutants

The single sgRNA was designed to target *GmPYL17*, *GmPYL18*, and *GmPYL19* based on the recognition and cutting principles of Cas9 protein in the CRISPR/Cas9 vector and referenced to whole-genome resequencing data of DN50 (Appendix A). In the three *GmPYLs* genes, the gRNA can target the second exon in the *GmPYL17*, *GmPYL18*, and *GmPYL19* genes, the gRNA targets the same sequence. In previous studies, the SNP at the distal end of the PAM sequence did not affect the recognition of the Cas9 protein [47]. This remarkable approach, which directs one gRNA to three unique genes simultaneously, greatly improves the efficiency of gene editing mutant generation. The designed gRNA can knock out *PYL17*, *PYL18,* and *PYL19* genes in soybean. The GmUbi3 promoter drives the Cas9 protein in the CRISPR vector, and gRNA is driven by the GmU6 promoter (Appendix A). The DN50 was selected as an explant for genetic transformation and directed mutagenesis because of its high transformation efficiency, short growth cycle, and suitability for growth chamber culture.

The *Agrobacterium*-mediated genetic transformation with soybean cotyledon node method was used, and 28 T_0_-transgenic lines were obtained (Appendix A). Firstly, the method of applying herbicide on the leaves (200 mg/L glyphosate) was used to identify successful transformation. Then, the existence of the T-DNA sequence was confirmed by PCR screening using specific primers of the cas9 gene and specific primers of the gRNA-pSC1 crossing region in transgene (Appendix A).

To identify CRISPR/Cas9-induced mutant genes, primers G13F/G13R, G03F/G03R, G06F/G06R were used to amplify DNA fragments covering the target regions of *GmPYL17*, *GmPYL18*, and *GmPYL19* genes from the genomic DNA of transgenic plants, respectively. Sanger sequencing revealed the amplified fragments, as well as two forms of mutation in the gene target region (Appendix A). We planted T_0_ plants in a greenhouse, collected seeds, and continued sowing the harvested seeds (T_1_ generation). When the unifoliolate leaves were fully unrolled, DNA was extracted, and three pairs of specific primers (G13F/G13R, G03F/G03R, G06F/G06R) were used to amplify the target fragments from the genomic DNA of transgenic plants and Sanger sequencing.

### 4.4. Genetic Markers Development

The sequence information of *GmPYL17*, *GmPYL18*, and *GmPYL19* mutation sites was obtained by Sanger sequencing (Beijing Tsingke Biology Co., Ltd., Beijing, China). Primers with an amplification product of about 800 bp were designed. PCR products with a single clear band and accurate band size detected by electrophoresis were sequenced. Further, the cleaved amplified polymorphic sequence (CAPS) primers were designed for more confirmation. The CAPS marker was developed based on the *GmPYL17*, *GmPYL18*, and *GmPYL19* sequence variations. After 37 °C for 4 h of enzymatic digestion, 4% agarose gel was used for electrophoresis.

### 4.5. Seed Treatment

Soybean seeds of wild type and mutants were soaked in 58 °C warm water and dried for 2 days. The well-dried seeds were selected for the germination experiment. Put two layers of filter paper at the bottom of the petri dish and add 25 soybean seeds to each plate. For the control experiment, we added 20 mL aseptic water; for the treatment group, we added 20 mL ABA solution (10 μM) and put it in dark culture at 25 °C, 60% relative humidity. The total number of seeds in each group (control and treatment) is more than 75 (3 Petri dishes per group). Take pictures every 6 h and record the number of germinated seeds in each group. After 60 h, calculate the germination rate of the treatment and control groups [65].

### 4.6. Analysis of Agronomic Characters of Mutant Plants

The T_2_ generation mutants and wild type plants were planted in the experimental field of Northeast Forestry University. The T_3_ strain was planted in the greenhouse of the College of Life Sciences of Northeast Forestry University (26 °C, light 14 h/dark 10 h). Place the pots in a non-porous sink and water from the bottom. Keep the surface of the soil slightly moist. Half of the soil used for planting was dug from the field, plus half of the grass charcoal soil. The phenotypes of T_3_ mutants and wild type were observed, and the flowering time was counted. The plant height and branch number of mutants and wild type plants at full maturity (R8) were recorded. After seed threshing, the 100-seed weight of mutant and wild type seeds was counted, and Student’s *t*-test was used to analyze the difference.

### 4.7. RNA Extraction, Library Construction, and Transcriptome Sequencing

Total mRNA was extracted from whole soybean seeds treated with water or ABA solution (10 μM) by the Trizol method. The RNA-seq data were obtained by high-throughput transcriptome sequencing, and the sequencing platform was Illumina Hiseq2500 at Shanghai Biozeron Biological Technology Co., Ltd. (Shanghai, China). Each sample produces 6.0-Gb sequencing data, sequenced by double-terminal (PE) sequencing with a reading length of 150 bp. Original data was obtained by sequencing to obtain high-quality clean reads by Fastp [66]. The clean reads were compared with the soybean reference genome (Phytozome v13, https://phytozome-next.jgi.doe.gov, accessed on 17 December 2022) using hisat2 [67]. After that, the gene’s FPKM value was quantitatively obtained using Stringtie [68]. The reads number of genes in each sample was obtained by Samtools [69]. Detection of differentially expressed genes (DEGs) using DEseq2 packages in the R package [70]. The criteria for screening differential genes were *p* < 0.05 and the fold change >2 (up-regulated or down-regulated). Gene Ontology was annotated by SoyBase (https://www.soybase.org/, accessed on 17 December 2022) [71]. The enrichment test was performed by the TBtools [72].

### 4.8. Quantitative Real-Time PCR (qRT-PCR) Analysis

RNA was extracted using the RNAprep pure Plant Kit (TIANGEN, Beijing, China). cDNA was synthesized using the PrimeScript RT reagent Kit (Takara, Beijing, China). qRT-PCR was performed using a LightCycler 480 SYBR Green I Master (Roche, Mannheim, Germany) in the Roche LightCycler480 system (Roche, Mannheim, Germany). A soybean housekeeping gene, Tubulin (Glyma.05G157300), was used as an internal control. The gene’s relative transcript levels were analyzed using the relative quantification method (2^−ΔΔCT^) [73].

## 5. Conclusions

In the present study, three genes, *GmPYL17*, *GmPYL18*, and *GmPYL19*, were altered using the CRISPR/Cas9 system, resulting in three homozygous mutants. The morphological and genetic effects on plant growth and development were observed. The mutant plants showed significant changes in the height and number of branches compared to wild types. Furthermore, at the seed germination stage, the mutants (*gmpyl17/19-1*) were insensitive to ABA treatment, which confirmed that the *GmAHG3* and *GmHONSU* were upregulated in the mutant lines and acted as positive regulators of seed germination. These findings provide a basis for the in-depth understanding of the ABA signal regulation network in soybean, especially the regulatory mechanism of seed germination.

## Figures and Tables

**Figure 1 ijms-23-16173-f001:**
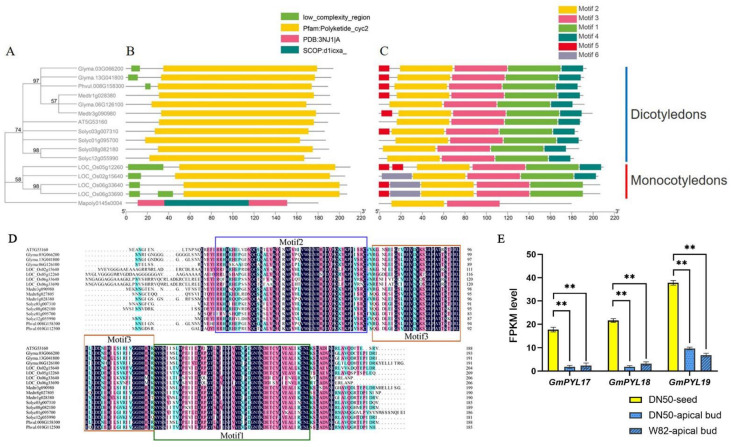
PYL8 genes in *Glycine max*, *Arabidopsis thaliana*, *Oryza sativa*, *Medicago truncatula*, *Solanum lycopersicum,* and *Marchantia polymorpha*. (**A**) Phylogenetic trees of six species constructed by the Maximum Likelihood were adopted using MEGA 10.0 software, with 1000 bootstrap replicates. (**B**) Analysis of the conserved domains PYLs. (**C**) The gene-conserved motifs of *PYL*s genes in six species are based on phylogenetic relationships. Monocotyledons and dicotyledons are represented by red and blue, respectively. (**D**) Amino acid sequence of motif1 (green box), motif2 (blue box), and motif3 (red box) of PYLs in six species. (**E**) FPKM of *GmPYL17*, *GmPYL18*, and *GmPYL19* in different tissues were determined from the transcriptome. The tissues are dry seed, shoot apical meristem of DN50, and Williams 82 (W82) plants. Values are means ± SD (n = 3). Student’s *t*-tests indicate significant differences in mean values relative to the mean value of seeds (** *p* < 0.01).

**Figure 2 ijms-23-16173-f002:**
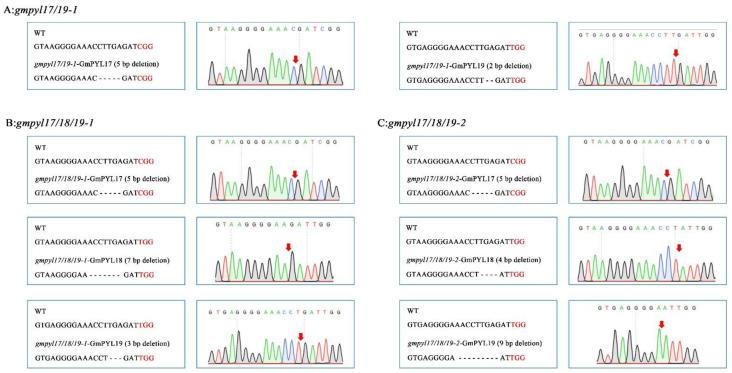
Homozygous of soybean *gmpyl17/18/19* generated by the CRISPR/Cas9 system in the T2 generation. Sequences of WT and mutant plants at target sites of *gmpyl17/19-1* (**A**), *gmpyl17/18/19-1* (**B**), and *gmpyl17/18/19-2* (**C**) mutants (**left**). Dashes indicate deleted nucleotides. The red color of nucleotides indicates PAM. The sequence peaks of the wild type (WT) and mutants at target sites (**right**). Red arrowheads indicate mutation locations.

**Figure 3 ijms-23-16173-f003:**
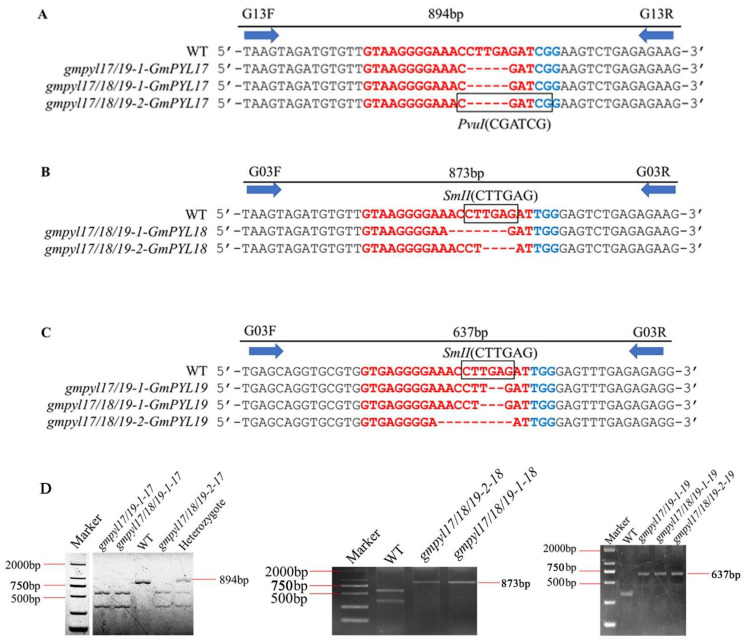
Genetic markers (CAPS) for distinguishing mutant and wild type alleles. (**A**) *GmPYL17* allele, (**B**) *GmPYL18* allele, (**C**) *GmPYL19* allele. Horizontal arrows indicate the molecular marker primer pair. The above number of continuous lines suggests the length of the significant PCR product, only the regions where mutations occur are shown here. Square boxes indicate restriction endonuclease *PvuI* and *SmII* digest sites. The red color of nucleotides indicates sgRNA; the blue color of nucleotides indicates PAM (**D**) The alleles in different types of mutations were identified by agarose gel electrophoresis.

**Figure 4 ijms-23-16173-f004:**
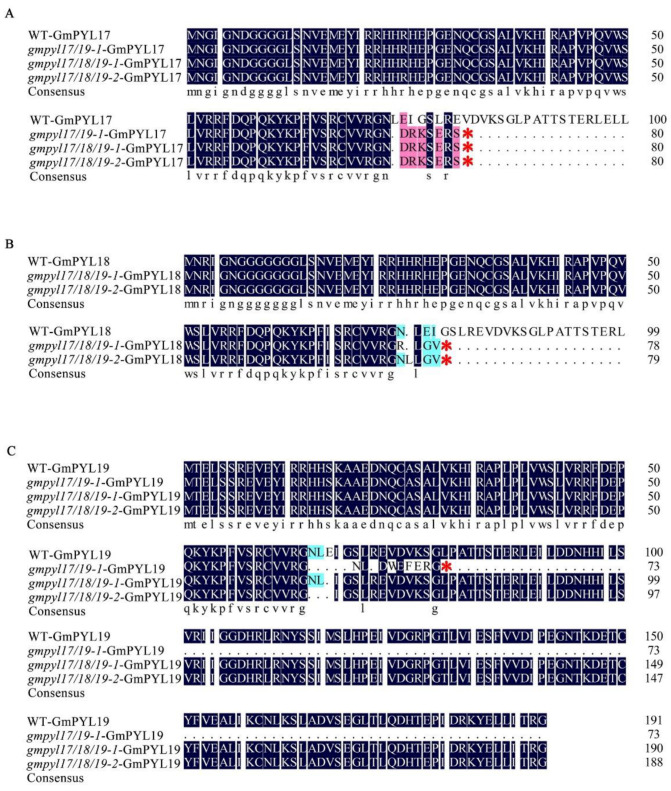
The predicted amino acid of *gmpyl17/19-1*, *gmpyl17/18/19-1*, *gmpyl17/18/19-2* mutations and wild type. (**A**) *GmPYL17* allele, (**B**) *GmPYL18* allele, (**C**) *GmPYL19* allele. The red “*” indicates the early termination of amino acids. Light blue, homology ≥ 50%; pink, homology ≥ 75%; blue, homology 100%.

**Figure 5 ijms-23-16173-f005:**
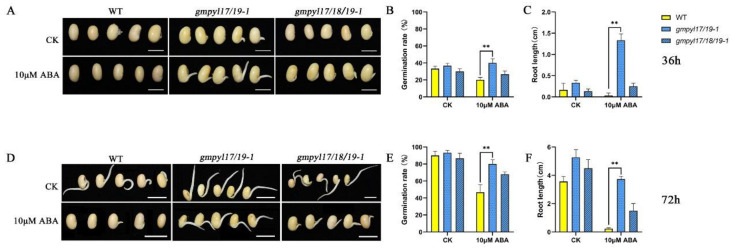
The germination rate of wild type, *gmpyl17/19-1,* and *gmpyl17/18/19-1* mutant seed. (**A**) Seed germination under distilled water control conditions. (**B**) Seed germination under 10 µM ABA treatment. (**C**,**D**) Germination rate of wild type (WT) and *gmpyl17/19-1*. (**E**,**F**) Germination rate of wild type (WT) and *gmpyl17/18/19-1*. Data points are averages of three biological replicates, n = 30. Error bars indicate standard deviations. (** *p* < 0.01).

**Figure 6 ijms-23-16173-f006:**
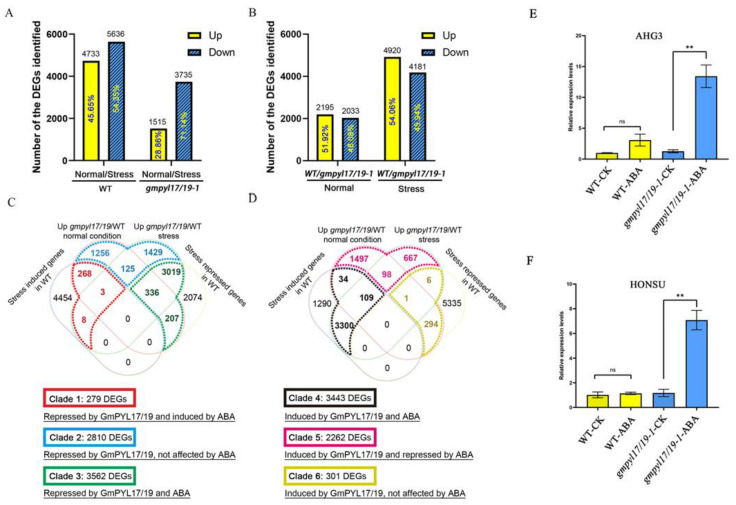
Overview of the RNAseq analysis and the six clades of DEGs regulated by *GmPYL17*/19. (**A**,**B**) Differentially expressed genes (DEGs) in comparing different lines and treatments. Normal in the plot indicates water, no ABA treatment, and stress indicates using 10 μM ABA stress treatment. *p*-value (adjusted *p*-value) < 0.05 and absolute log2 ratio > 1 were used for significant differential expression cutoffs. (**C**,**D**) Venn diagram of the DEGs in different genotypes under normal and ABA stress. Based on their expression patterns, the DEGs affected by the interaction of *GmPYL17*/19 and ABA stress and the *GmPYL17*/19 alone can be grouped into six different clades. The upregulated DEGs in the *gmpyl17/19-1* mutant panel (**C**) were grouped into Clades 1–3 (279 DEGs, 2810 DEGs, and 3562 DEGs) based on how they responded to ABA stress (**C**). The upregulated DEGs in the *gmpyl17/19* mutant panel (**D**) were grouped into Clades 4–6 (3443 DEGs, 2262 DEGs, and 301 DEGs) based on how they responded to ABA stress (**D**). The colors in the same clade were boxed the same color as in the Venn diagram. (**E**) Transcriptional levels of *AHG3* in WT and the mutant of soybean *gmpyl17/19-1* under greenhouse conditions (14 h light/10 h dark). (**F**) Transcriptional levels of *HONSU* in WT and the mutant of soybean *gmpyl17/19-1* under greenhouse conditions. The values shown are relative to the control gene *TUB* and represent the means ± standard error of the mean (s.e.m.) of three biological replications. Significant differences were identified by Student’s *t*-test (** *p* < 0.01; ns *p* > 0.05).

**Figure 7 ijms-23-16173-f007:**
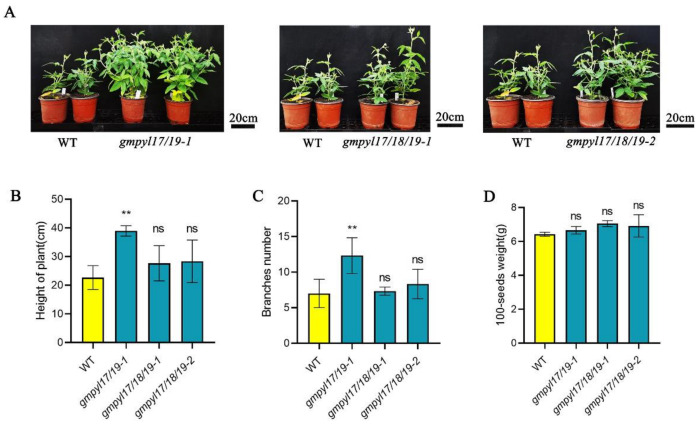
Crop traits of the *GmPYLs* mutant. (**A**) Phenotypes of WT and mutants (*gmpyl17/18-1*, *gmpyl17/18/19-1*, *gmpyl17/18/19-2*) under greenhouse conditions. (**B**) Statistics analysis of the height of the plant of WT and the mutant. (**C**) Statistics analysis of the branch number of WT and the mutant. (**D**) 100-seeds weight of WT and the quadruple mutant. All data are presented as mean ± s.e.m. (n = 10 plants). Significant differences were identified by Student’s *t*-test (** *p* < 0.01, ns *p* > 0.05).

## Data Availability

RNA-seq data download from PRJCA000740.

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
