# Peer review of "CRISPR/Cas9-Mediated Multiple Knockouts in Abscisic Acid Receptor Genes Reduced the Sensitivity to ABA during Soybean Seed Germination"

_ijms, 2022, doi:10.3390/ijms232416173_

Round 1
Reviewer 1 Report
The manuscript by Zhang et al. is devoted to an important problem: the study of the mechanisms of ABA signal transduction and their influence on the germination of soybean seeds. This work contains very interesting results, but I have some comments to the manuscript:
1. P. 1-2, lines 41-70: It is important to better structure the information for easier understanding by the reader. Perhaps a graphical diagram illustrating this section will help.
2. P. 3, line 111: Please, give an explanation of “DN50” because it appears in the text for the first time.
3. P. 3, Figure 1E:
Please, give an explanation of “W82” in the legend.
I don't see SDs on the graph.
4. P. 8, Figure 5: Please, add “** - p Ë‚ …” to the legend.
5. P. 11, Figure 7: “The mutant showed higher plant height and more branches than WT.” is a strange name of the figure.
6. P. 11-12, lines 297-319: It would be appropriate to add here literature data on how mutants with reduced sensitivity to ABA survive and function under drought stress conditions. If they die or significantly reduce productivity, their germination does not make sense.
7. P. 3, line 419-424: How did you water the plants in the greenhouse during the experiment? What soil moisture was maintained? What was the type and composition of the soil? This information should be added.
Author Response
Response to reviewer
We thank you for the critical comments and helpful suggestions. We have taken all these comments and suggestions into account as follows,and all the response use red color fonts , all changes in main text are highlighted in yellow:
- P. 1-2, lines 41-70:It is important to better structure the information for easier understanding by the reader. Perhaps a graphical diagram illustrating this section will help.
Thank you for your comments, We rewrite the sentences and modified it for easier understanding.
- P. 3, line 111:Please, give an explanation of “DN50” because it appears in the text for the first time.
Thanks for the reminder, we have modified this description in the text.
- P. 3, Figure 1E:
Please, give an explanation of “W82” in the legend.
Thank you for your comments, we have modified this description in the legend.
I don't see SDs on the graph.
Thanks for the reminder, we have redraw Figure 1E, and add error bars.
- P. 8, Figure 5: Please, add “** - p Ë‚ …” to the legend.
Thank you for your comments, we have add “** - p Ë‚ 0.01” in the legend.
- P. 11, Figure 7: “The mutant showed higher plant height and more branches than WT.” is a strange name of the figure.
Thanks for the reminder. Our original writing did give rise to ambiguity, and we have now revised the title according to comments as follows: Crop traits of the GmPYLs mutant.
- P. 11-12, lines 297-319: It would be appropriate to add here literature data on how mutants with reduced sensitivity to ABA survive and function under drought stress conditions. If they die or significantly reduce productivity, their germination does not make sense.
Thank you for your comments, and thank you for your recognition of our work. Many papers reveal that overexpression of ABA receptor gene can increase drought resistance, our experimental studies on drought resistance are ongoing, and will be described in the next article. Our ABA-insensitive mutant has a higher plant biomass (as reflected by more branches and more seeds) under normal conditions, which will lead to increased soybean yield.
- P. 3, line 419-424: How did you water the plants in the greenhouse during the experiment? What soil moisture was maintained? What was the type and composition of the soil? This information should be added.
Thank you for your comments, we add the method of water the plants, soil moisture and composition of the soil as follow: We place the pots in a non-porous sink and water from the bottom. Keep the surface of the soil slightly moist. The soil used for planting was one-half of soil dug from the field, plus one-half of grass charcoal soil.
Reviewer 2 Report
Major comments:
The manuscript by Zhang et al describes the gene mutation by CRISPR-Cas9 and bioinformatic analysis of soybean PYL17-PYL19 genes. This is an interesting piece of work on a group of ABA receptor genes which is analyzed for the first time in this plant and some of their mutant phenotypes were here described.
ABA is known to be involved in drought stress response and some ABA-insensitive or ABA-hypersensitive mutants have a drought tolerant phenotype. However, the authors leave us with an important question unanswered: are these Gmpyl double or triple mutants drought tolerant or not? It’s not a difficult nor time-consuming (if plantlets are used) experiment to make.
Minor comments:
Line 15: replace regulated with regulate.
Lines 95-97: rephrase it is confusing; I suggest: Three soybean homologue genes Glyma.13G041800, Glyma.03G066200, and Glyma.06G126100 were named GmPYL17, GmPYL18, and GmPYL19 and displayed homology to AtPYL8 from Arabidopsis.
Line 151: should say: The deduced protein sequence.
In Figure 3C the SmII should be in italics.
Line 319: add a period at the end.
Lines 320-322: rephrase it, is unclear. I suggest: In gmpyl17/19-1 mutant the mutation type of the GmPYL19 allele is early termination of translation, whereas in gmpyl17/18/19-1 mutant the mutation type of GmPYL19 321 allele is an amino acid deletion.
Author Response
Response to reviewer
Thank you for your comments and suggestion concerning our manuscript. The comments and suggestions are all valuable and very helpful for revising and improving our paper, as well as the important guiding significance to our researches. We have studied comments carefully and have made correction which we hope meet with approval. We have taken all these comments and suggestions into account as follows,and all the response use red color fonts , all changes in main text are highlighted in yellow:
Major comments:
The manuscript by Zhang et al describes the gene mutation by CRISPR-Cas9 and bioinformatic analysis of soybean PYL17-PYL19 genes. This is an interesting piece of work on a group of ABA receptor genes which is analyzed for the first time in this plant and some of their mutant phenotypes were here described.
ABA is known to be involved in drought stress response and some ABA-insensitive or ABA-hypersensitive mutants have a drought tolerant phenotype. However, the authors leave us with an important question unanswered: are these Gmpyl double or triple mutants drought tolerant or not? It’s not a difficult nor time-consuming (if plantlets are used) experiment to make.
Thank you so much for a very important comments and thank you for your recognition of our work! We focused on seed germination, drought tolerant is ongoing experiments. We will publish drought stress tolerance in other paper.
Minor comments:
Line 15: replace regulated with regulate.
Thanks for the reminder, we have modified this description in the text.
Lines 95-97: rephrase it is confusing; I suggest: Three soybean homologue genes Glyma.13G041800, Glyma.03G066200, and Glyma.06G126100 were named GmPYL17, GmPYL18, and GmPYL19 and displayed homology to AtPYL8 from Arabidopsis.
Thanks for the reminder. Our original writing did give rise to ambiguity, and we have now revised it according to comments as follows: Three soybean homologue genes Glyma.13G041800, Glyma.03G066200, and Glyma.06G126100 were named GmPYL17, GmPYL18, and GmPYL19 and displayed homology to AtPYL8 from Arabidopsis.
Line 151: should say: The deduced protein sequence.
Thanks for the reminder, we have modified this description in the text.
In Figure 3C the SmII should be in italics.
Thanks for the reminder, we have changed the font in Figure 3C.
Line 319: add a period at the end.
Thanks for the reminder, we have modified this punctuation in the text.
Lines 320-322: rephrase it, is unclear. I suggest: In gmpyl17/19-1 mutant the mutation type of the GmPYL19 allele is early termination of translation, whereas in gmpyl17/18/19-1 mutant the mutation type of GmPYL19 allele is an amino acid deletion.
Thanks for the reminder. Our original writing did give rise to ambiguity, and we have now revised it according to the reviewers' comments as follows: In gmpyl17/19-1 mutant the mutation type of the GmPYL19 allele is early termination of translation, whereas in gmpyl17/18/19-1 mutant the mutation type of GmPYL19 allele is an amino acid deletion.